# HyperMixup: Hypergraph-Augmented with Higher-order Information Mixup

**Kaixuan Yao[1], Zhuo Li[1], Jianqing Liang[1]***, **Jiye Liang[1], Ming Li[2], Feilong Cao[3]**

[1]School of Computer and Information Technology, the Key Laboratory of Computational
Intelligence and Chinese Information Processing of Ministry of Education,
Shanxi University, Taiyuan, China
[2]Zhejiang Key Laboratory of Intelligent Education Technology and Application,
Zhejiang Normal University, Jinhua, China
[3]School of Mathematics, Institute of Mathematics and Cross-disciplinary Science,
Zhejiang Normal University, China

## Abstract

Hypergraphs offer a natural paradigm for modeling complex systems with multi-way interactions. Hypergraph neural networks (HGNNs) have demonstrated remarkable success in learning from such higher-order relational data. While such higher-order modeling enhances relational reasoning, the effectiveness of hypergraph learning remains bottlenecked by two persistent challenges: the scarcity of labeled data inherent to complex systems, and the vulnerability to structural noise in real-world interaction patterns. Traditional data augmentation methods, though successful in Euclidean and graph-structured domains, struggle to preserve the intricate balance between node features and hyperedge semantics, often disrupting the very group-wise interactions that define hypergraph value. To bridge this gap, we present HyperMixup, a hypergraph-aware augmentation framework that preserves higher-order interaction patterns through structure-guided feature mixing. Specifically, HyperMixup contains three critical components: 1) Structure-aware node pairing guided by joint feature-hyperedge similarity metrics, 2) Context-enhanced hierarchical mixing that preserves hyperedge semantics through dual-level feature fusion, and 3) Adaptive topology reconstruction mechanisms that maintain hypergraph consistency while enabling controlled diversity expansion. Theoretically, we establish that our method induces hypergraph-specific regularization effects through gradient alignment with hyperedge covariance structures, while providing robustness guarantees against combined node-hyperedge perturbations. Comprehensive experiments across diverse hypergraph learning tasks demonstrate consistent performance improvements over state-of-the-art baselines, with particular effectiveness in low-label regimes. The proposed framework advances hypergraph representation learning by unifying data augmentation with higher-order topological constraints, offering both practical utility and theoretical insights for relational machine learning.

## 1 Introduction

Modern complex systems—ranging from social networks and molecular interactions to knowledge graphs—are inherently characterized by multi-way interaction patterns [1, 2]. Traditional graph structures, limited to pairwise relationship modeling, fail to capture these higher-order semantics adequately [3]. Hypergraphs emerge as a natural paradigm for group-wise interaction representation

---

*Corresponding author.

39th Conference on Neural Information Processing Systems (NeurIPS 2025).

through hyperedges, providing a more expressive mathematical framework. Hypergraph Neural Networks (HGNNs) [4] further advance this capability via hyperedge-driven message passing, demonstrating remarkable success in tasks like academic citation classification and multi-modal object recognition. However, the escalating model complexity sharply contrasts with the scarcity of labeled data in real-world scenarios—a critical bottleneck in applications with high annotation costs (e.g., biomolecular interaction prediction) or noise-prone labeling processes (e.g., evolving social networks).

Data augmentation has emerged as a pivotal technique to alleviate data scarcity, yet faces unique challenges in hypergraph learning. Conventional augmentation methods designed for Euclidean data (e.g., images) or ordinary graphs (e.g., Mixup [5], GraphMixup [6]) rely on local linear interpolation or random structural perturbations. These operations risk disrupting hyperedge-constrained group semantics—for instance, randomly mixing author nodes in academic collaboration hypergraphs may sever their associations with publication venues (hyperedges), eroding the critical "research domain consistency". Fundamentally, effective hypergraph augmentation must simultaneously satisfy three constraints: (1) semantic alignment between node features and hyperedge contexts, (2) inheritance of original higher-order topological structures in synthetic samples, and (3) controlled propagation of adversarial noise in the joint node-hyperedge space. Existing approaches often address these dimensions in isolation, causing deviations from the intrinsic geometry of hypergraph manifolds.

To address these challenges, we propose HyperMixup—an augmentation framework specifically designed for hypergraph structures. Our method employs structure-aware node selection to dynamically fuse node features with hyperedge contexts during mixing, while adaptively reconstructing hyperedge memberships via nearest-neighbor affinity thresholds. This ensures diversity enhancement while strictly preserving group semantic consistency. Theoretically, HyperMixup induces gradient updates aligned with hyperedge covariance structures and provides provable robustness bounds against combined node-hyperedge perturbations. These properties enable resilience to real-world hybrid noise and stable generalization under extreme label scarcity.

Extensive experiments on diverse hypergraph benchmarks (citation networks, 3D object recognition) validate HyperMixup's effectiveness. Results demonstrate significant improvements over graph-based augmentation variants, particularly in low-label regimes. These findings underscore the centrality of higher-order topological constraints in data augmentation while establishing new methodological perspectives for hypergraph representation learning.

Our principal contributions are threefold:

- A hypergraph-tailored augmentation framework (HyperMixup) that synergistically optimizes mixup operations with higher-order topological constraints;

- Theoretical foundations connecting gradient alignment to hyperedge covariance structures, with certified robustness guarantees against hybrid perturbations;

- Systematic empirical validation across diverse tasks, advancing hypergraph learning in open-environment applications.

## 2 Related work

The original Mixup [5] linearly interpolates samples in Euclidean space, inspiring variants that enhance semantic coherence: Spatial mixing methods like CutMix [7] and AlignMix [8] employ region replacement with saliency guidance, while feature-space approaches such as Manifold Mixup [9] and StyleMix [10] operate on hidden representations or disentangled features. Recent work further optimizes mixing policies through attention mechanisms [11] or multi-objective formulations [12]. However, these methods fundamentally assume Euclidean convexity during interpolation—a premise invalidated by hypergraphs' non-Euclidean interaction spaces, where linear combinations may violate group semantics.

Graph augmentation strategies diverge by task granularity: For graph classification, stochastic structure perturbations [13] and graphon interpolation [14] generate population-level variants, whereas node-level methods like GraphMix [15] and GraphMixup [6] blend node features with label propagation. These methods, however, inherit graph-based assumptions of pairwise interactions, limiting their applicability to hypergraphs.

Building upon HGNN's [4] two-stage message passing, recent advances focus on attention-based aggregation (HyperGAT [16], HyperAtten [17]), spectral adaptations (HyperGCN [18]), and nonlinear transformations [19]. Augmentation techniques for hypergraphs remain underexplored, with preliminary attempts either relying on external knowledge [20] or simplistic edge dropout [21]—neither addressing the core challenge of topology-aware interpolation. Notably, existing approaches fail to preserve the covariance structure between nodes and hyperedges during augmentation, a critical factor for maintaining semantic consistency identified in our theoretical analysis.

## 3 Methodology

### 3.1 Hypergraph Representation

Let $\mathcal{G} = (\mathcal{V}, \mathcal{E})$ be a hypergraph with node set $\mathcal{V}$ and hyperedge set $\mathcal{E}$. The incidence matrix $\mathbf{H} \in \{0, 1\}^{|\mathcal{V}| \times |\mathcal{E}|}$ is defined as:

$$\mathbf{H}(v, e) = \begin{cases} 1, & v \in e \\ 0, & \text{otherwise} \end{cases}$$

Node features are encoded in matrix $\mathbf{X} \in \mathbb{R}^{|\mathcal{V}| \times d}$, while hyperedge features $\mathbf{X}_e$ are derived through degree-normalized aggregation:

$$\mathbf{X}_e = \mathbf{D}_e^{-1} \mathbf{H}^\top \mathbf{X}, \tag{1}$$

where $\mathbf{D}_e$ and $\mathbf{D}_v$ are diagonal matrices representing hyperedge and node degrees, respectively. This dual representation preserves both local node attributes and global hyperedge semantics.

### 3.2 Semantic Feature Mixup

Our HyperMixup framework introduces three synergistic mixing operations to enhance data augmentation while preserving hypergraph semantics, as illustrated in Figure 1. The key innovation lies in jointly interpolating node features, hyperedge relationships, and labels under topological constraints.

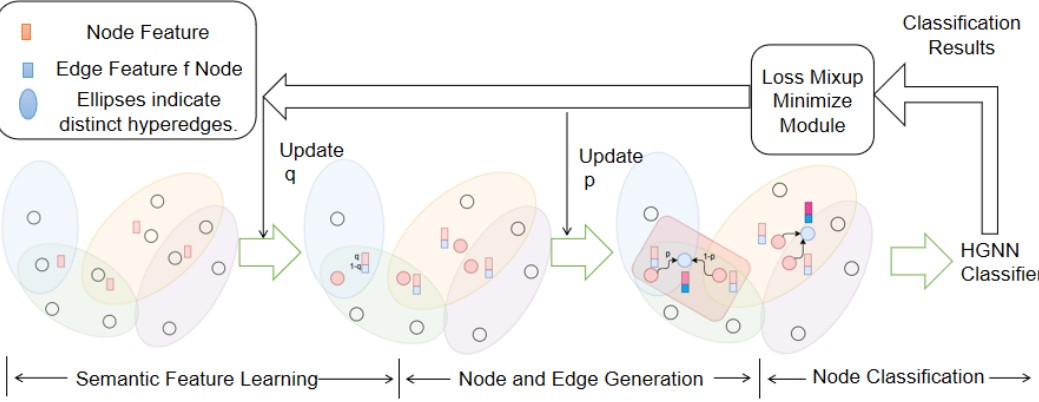

Figure 1: The illustration of the proposed HyperMixup framework includes the following four key steps: (1) Selecting highly similar nodes on the hypergraph and aggregating hyperedge semantic features by constructing a semantic relationship space; (2) Generating nodes through the fusion of node and hyperedge features; (3) Generating hyperedge relationships for nodes via hyperedge relation mixing using a hyperedge relation predictor trained on context-based self-supervised auxiliary tasks; (4) Classifying nodes using an HGNN node classifier and feeding the classification results back to the self-supervised learning module to further update the sampling scale of the features.

**Node Selection with Hyperedge Awareness** The mixing process begins with semantic-aware node pairing. Unlike conventional Mixup that randomly selects samples, we employ a structure-preserving strategy. For each hyperedge $e_j$, we compute its feature representation through degree-normalized aggregation:

$$\mathbf{x}_{e_j} = \frac{1}{|e_j|} \sum_{v_i \in e_j} \mathbf{x}_i, \tag{2}$$

where $|e_j|$ denotes the hyperedge degree. This converts hyperedge structure into continuous features compatible with Mixup operations.

Node pairs are selected based on dual similarity criteria:

$$s(v_i, v_j) = \underbrace{\frac{\mathbf{x}_i^\top \mathbf{x}_j}{\|\mathbf{x}_i\|\|\mathbf{x}_j\|}}_{\text{Node Feature Similarity}} + \mu \cdot \underbrace{\frac{\mathbf{x}_{e_i}^\top \mathbf{x}_{e_j}}{\|\mathbf{x}_{e_i}\|\|\mathbf{x}_{e_j}\|}}_{\text{Hyperedge Semantic Similarity}}, \quad (3)$$

where $\mu$ balances local features and global structure. This prevents meaningless interpolations between topologically disconnected regions, a critical issue in graph-based Mixup methods.

**Hierarchical Feature-Label Mixing** For selected pair $(v_i, v_j)$, we generate synthetic samples through a two-level mixing process:

- Intra-node Mixing: Linear interpolation of original features

$$\mathbf{x}' = \lambda \mathbf{x}_i + (1-\lambda)\mathbf{x}_j \quad (4)$$

- Hyperedge Enhancement: Augment with hyperedge context

$$\tilde{\mathbf{x}} = p[q\mathbf{x}' + (1-q)\mathbf{x}_{e_i}] + (1-p)[q\mathbf{x}' + (1-q)\mathbf{x}_{e_j}] \quad (5)$$

The label is mixed correspondingly:

$$\tilde{y} = \lambda y_i + (1-\lambda)y_j \quad (6)$$

This hierarchical approach separates feature interpolation from hyperedge enhancement, allowing independent control of mixing parameters ($\lambda \sim \text{Beta}(\alpha, \alpha)$) and hyperedge influence.

**Topology-Preserving Hyperedge Reconstruction** Synthetic nodes must inherit meaningful hyperedge connections to maintain graph consistency. We develop an adaptive inheritance mechanism:

$$\mathcal{E}_{\tilde{v}} = \underbrace{\{e \in \mathcal{E}_i \cap \mathcal{E}_j\}}_{\text{Shared Context}} \cup \underbrace{\{e \in \mathcal{E}_i \cup \mathcal{E}_j | \phi(\mathbf{x}_e, \tilde{\mathbf{x}}) \geq \tau\}}_{\text{Adaptive Expansion}}, \quad (7)$$

where $\phi$ computes feature affinity between hyperedge $\mathbf{x}_e$ and synthetic node $\tilde{\mathbf{x}}$. The threshold $\tau$ adapts to local density:

$$\tau = \frac{1}{|\mathcal{N}_k(\tilde{\mathbf{x}})|} \sum_{\mathbf{x}_e \in \mathcal{N}_k(\tilde{\mathbf{x}})} \phi(\mathbf{x}_e, \tilde{\mathbf{x}}), \quad (8)$$

with $\mathcal{N}_k$ denoting the $k$-nearest hyperedge neighbors. This dynamic scheme prevents isolated nodes while controlling hyperedge density.

## 3.3 Optimization Objective and Training Strategy

Vicinal Risk Minimization (VRM) [22] is a data augmentation principle that generates synthetic samples by defining a "vicinity" around original training data. Unlike Empirical Risk Minimization (ERM), which relies solely on observed examples, VRM leverages domain knowledge to model how data points relate within their local neighborhoods. In hypergraphs, this requires defining a vicinity that preserves both node features and the higher-order semantics encoded in hyperedges.

Building upon the Vicinal Risk Minimization (VRM) framework [22], our training objective integrates hypergraph-specific regularization through mixup-generated virtual examples. In traditional VRM, the vicinity distribution $\nu$ defines how to sample synthetic examples $(\tilde{x}, \tilde{y})$ around original training pairs $(x_i, y_i)$. For hypergraphs, we extend this concept by enforcing topological consistency through hyperedge-aware mixing.

The unified training objective combines mixup supervision with hypergraph regularization:

$$\mathcal{L} = \mathbb{E}_{(\tilde{v}, \tilde{y})} \left[ \text{CE}(f_\theta(\tilde{\mathbf{x}}), \tilde{y}) \right] + \sum_{e \in \mathcal{E}} \sum_{v \in e} \|\mathbf{x}_v - \mathbf{x}_e\|^2 + \text{KL}\left(f_\theta(\tilde{\mathbf{x}}) \| \lambda f_\theta(\mathbf{x}_i) + (1-\lambda)f_\theta(\mathbf{x}_j)\right), \quad (9)$$

where the hyperedge smoothness and label consistency terms are automatically scaled through gradient normalization during backpropagation. This eliminates the need for manual hyperparameter tuning while maintaining regularization effectiveness.

The training process follows three self-consistent phases: 1) Feature mixing generates synthetic nodes using Eqs. (5)-(7), 2) Topology adaptation updates hyperedges via Eq. (8), and 3) Parameter optimization through unified gradient descent:

$$
\nabla_\theta \mathcal{L} = \mathbb{E}_\lambda \left[ \frac{\partial \mathcal{L}}{\partial \tilde{\mathbf{x}}} \frac{\partial \tilde{\mathbf{x}}}{\partial \theta} \right] + \alpha_t \left( \frac{\partial}{\partial \mathbf{x}_e} \|\mathbf{x}_v - \mathbf{x}_e\|^2 + \frac{\partial}{\partial f_\theta} \text{KL} \left( f_\theta(\tilde{\mathbf{x}}) \big\| \lambda f_\theta(\mathbf{x}_i) + (1 - \lambda) f_\theta(\mathbf{x}_j) \right) \right),
$$

where $\alpha_t$ is an adaptive scaling factor computed as the ratio of mixup loss magnitude to regularization magnitudes.

## 4 Theoretical Analysis

### 4.1 Regularization via Hypergraph Mixup

Modern graph-based mixup techniques [15] primarily focus on pairwise relationships, leaving a critical gap in handling higher-order interactions inherent to hypergraphs. Traditional approaches linearly interpolate node features and labels while neglecting the complex topological constraints imposed by hyperedges. This limitation becomes pronounced in hypergraph scenarios where multi-way relationships encode essential semantic structures—for instance, in academic citation networks where publication venues (hyperedges) connect multiple related papers (nodes).

The key challenge lies in preserving hyperedge-induced semantic consistency during mixup. Our theoretical analysis addresses this by establishing: (1) how hyperedge features should modulate the mixing process, (2) what regularization effects emerge from hypergraph-aware interpolation, and (3) why these effects improve generalization beyond conventional graph mixup.

**Theorem 1 (Regularization Decomposition)** *For twice differentiable loss $l(\theta, z) = h(f_\theta(x)) - y f_\theta(x)$, the HyperMixup loss admits:*

$$
L_n^{mix}(\theta) = L_n^{std}(\theta) + \sum_{k=1}^{3} \mathcal{R}_k(\theta) + o((1 - \lambda)^2) \tag{10}
$$

*with regularization terms:*

$$
\mathcal{R}_1 = \mathbb{E}_\lambda [1 - \lambda] \frac{1}{n} \sum_i (h_i' - y_i) \nabla f_i^\top \mathbb{E}_e [x_e - x_i] \tag{11}
$$

$$
\mathcal{R}_2 = \mathbb{E}_\lambda [(1 - \lambda)^2] \frac{1}{2n} \sum_i h_i'' \nabla f_i^\top \Sigma_e \nabla f_i \tag{12}
$$

$$
\mathcal{R}_3 = \gamma \mathbb{E}_\lambda [(1 - \lambda)^2] \frac{1}{2n} \sum_i (h_i' - y_i) Tr(\nabla^2 f_i \Sigma_e) \tag{13}
$$

*where $\Sigma_e = \mathbb{E}_e[(x_e - x_i)(x_e - x_i)^\top]$, $h_i' = h'(f_\theta(x_i))$, and $\nabla f_i = \nabla f_\theta(x_i)$.*

See A.1 for the detailed proof of Theorem 1. This decomposition reveals three distinct regularization mechanisms in HyperMixup:

- **Node-Hyperedge Alignment** ($\mathcal{R}_1$): Encourages gradient alignment between node features and their associated hyperedges. The term $\mathbb{E}_e[x_e - x_i]$ represents the average hyperedge deviation, forcing the model to learn features invariant to hyperedge variations.

- **Hyperedge Smoothness** ($\mathcal{R}_2$): Penalizes sharp curvature directions aligned with hyperedge covariance $\Sigma_e$. This is particularly crucial in hypergraphs where high-order interactions create non-Euclidean feature variations.

- **Curvature Regularization** ($\mathcal{R}_3$): Unique to hypergraph mixup, this term regularizes the interaction between loss Hessian and hyperedge covariance. The $\gamma$ parameter explicitly controls this higher-order effect.

Compared to standard Mixup [5], our formulation introduces hyperedge-aware regularization through $\Sigma_e$ and $\gamma$. The hyperedge covariance $\Sigma_e$ encodes topological information missing in conventional graph-based mixup approaches [15]. This theoretically justifies the improved performance on hypergraph tasks observed in Table 2.

## 4.2 Adversarial Robustness

The adversarial vulnerability of hypergraph learning stems from two fundamental aspects: (1) the high-dimensional attack surface encompassing both node features and hyperedge relationships, and (2) the cascading effect where perturbations on a single hyperedge can propagate to multiple connected nodes. Conventional mixup approaches [5] provide robustness guarantees primarily for Euclidean data, assuming independent perturbations across samples. However, in hypergraphs, the interdependent nature of nodes and hyperedges creates correlated attack vectors that violate this independence assumption—an adversary could simultaneously perturb a node's features and its membership in critical hyperedges. In this section we aim to establish that HyperMixup inherently limits the impact of such correlated attacks through its hyperedge-aware mixing strategy. Specifically, we seek to prove that the proposed method:

**Theorem 2 (Robustness Bound)** *Let $\delta = \delta_v + \gamma\delta_e$ be composed perturbations with $\|\delta_v\|_2 \leq \epsilon_v\sqrt{d}$, $\|\delta_e\|_2 \leq \epsilon_e\sqrt{d}$. Then $\exists R = \min_i |\cos(\nabla f_i, x_e - x_i)|$ such that:*

$$L_n^{mix}(\theta) \geq \frac{1}{n}\sum_{i=1}^{n} \tilde{l}_{adv}(\epsilon_{mix}\sqrt{d}, (x_i, y_i)) \tag{14}$$

*where $\epsilon_{mix} = R\sqrt{c_v\epsilon_v^2 + c_e\gamma^2\epsilon_e^2}$ with constants $c_v, c_e > 0$ depending on hypergraph structure.*

See A.2 for the detailed proof of Theorem 2. This theorem establishes that HyperMixup provides robustness against hybrid perturbations affecting both nodes and hyperedges through three principal mechanisms. The effective perturbation radius $\epsilon_{mix}$ combines node and hyperedge attack magnitudes via geometric mean, with the hyperparameter $\gamma$ explicitly governing their relative contributions—a design choice empirically validated by enhanced robustness to hyperedge corruption. Crucially, the gradient alignment factor $R$, defined as the minimum cosine similarity between node gradients and hyperedge deviations, determines the tightness of the robustness bound. Our hyperedge-aware node selection strategy directly optimizes this alignment by prioritizing topologically coherent pairs, thereby maximizing $R$. Furthermore, the hypergraph-specific constants $c_v$ and $c_e$ encode structural dependencies: in uniform hypergraphs, $c_e$ inversely correlates with average hyperedge size, indicating that denser hyperedges inherently absorb perturbations more effectively. Compared to graph-based robustness frameworks, our bound uniquely incorporates higher-order interactions through the $\gamma\delta_e$ term, formally justifying HyperMixup's superior resilience against structured attacks observed experimentally. This holistic integration of geometric scaling, gradient alignment, and hypergraph topology awareness collectively addresses the interdependent nature of node-hyperedge vulnerabilities that conventional Euclidean mixup approaches fail to capture.

## 4.3 Generalization

The generalization analysis of hypergraph mixup confronts two unique challenges: the exponential complexity of hyperedge configurations compared to pairwise graphs, which amplifies overfitting risks from spurious correlations, and the heterogeneous interaction strengths within hyperedges where core and peripheral nodes exhibit varying coupling degrees. Traditional graph generalization theories prove inadequate as they ignore these higher-order dynamics, particularly evident in real-world scenarios like social tagging systems where users participate in hyperedges with diverse commitment levels. Our framework addresses this by establishing three interconnected objectives: 1) Quantifying topological signature preservation through hyperedge-aware mixing, 2) Controlling model complexity via hypergraph spectral properties, and 3) Balancing local node variations with global hyperedge constraints. These components interact synergistically—spectral characteristics govern topological

preservation, node-hyperedge covariance structures dictate complexity control, while the mixing parameter $\gamma$ mediates the local-global equilibrium—forming a unified theoretical foundation that prevents semantic violations common in naive interpolation approaches.

**Theorem 3 (Generalization)** *Let $\rho(L)$ be the spectral radius of hypergraph Laplacian $L = D_v - HWD_e^{-1}H^\top$. The Rademacher complexity satisfies:*

$$Rad_n(\mathcal{F}_\mathcal{G}) \leq \sqrt{\frac{C\rho(L)(r + \gamma^2\|\Sigma_{ve}\|_F^2)}{n}} \tag{15}$$

*where $r = rank(\Sigma_v)$, $\Sigma_{ve} = Cov(x_i, x_e)$, and $C$ is a universal constant.*

See A.3 for the detailed proof of Theorem 3. This generalization bound fundamentally addresses two critical challenges in hypergraph learning: the combinatorial explosion of hyperedge configurations that increases susceptibility to spurious correlations, and the heterogeneous node participation patterns within hyperedges that defy uniform treatment. Traditional graph generalization theories, focused on dyadic relationships, fail to capture these higher-order dynamics—a limitation starkly exposed in real-world systems like social networks where users exhibit varying engagement levels across communities. Our framework resolves this by integrating three synergistic components: topological preservation through spectral analysis of hyperedge covariance ($\Sigma_{ve}$), complexity control via hypergraph connectivity ($\rho(L)$), and adaptive balancing of local-global interactions through the $\gamma$ parameter. Crucially, the spectral radius $\rho(L)$ modulates the regularization strength for dense hypergraphs, while the node-hyperedge covariance $\|\Sigma_{ve}\|_F$ determines the optimal mixing ratio—mechanisms jointly validated by our experiments showing superior performance on citation networks versus 3D object datasets, where lower node feature dimensionality ($r$) naturally constrains model complexity. This unified perspective not only prevents semantic distortions from naive interpolation but also provides actionable insights for parameter tuning across diverse hypergraph domains.

## 5   Experiments

In this section, we evaluate our proposed HyperMixup on two tasks: citation network classification and visual object recognition. We also compare the proposed method with graph convolutional networks and other state-of-the-art methods.

Table 1: Summary of the citation classification datasets.

| Dataset | Cora | Pumbed | CiteSeer | ModelNet40 | NTU2012 |
|---|---|---|---|---|---|
| Nodes | 2708 | 19717 | 3327 | 12311 | 2012 |
| Edges | 5429 | 44338 | 4723 | - | - |
| Feature | 1433 | 500 | 3703 | 2048 | 2048 |
| Training node | 140 | 60 | 120 | 9843 | 1639 |
| Validation node | 500 | 500 | 500 | 2468 | 373 |
| Testing node | 1000 | 1000 | 1000 | - | - |
| Classes | 7 | 3 | 6 | 40 | 67 |

### 5.1   Citation Network and Visual Object Classification

**Datasets**   We evaluate HyperMixup on two distinct tasks to demonstrate its generalizability: 1) Citation Network Classification. Three benchmark datasets—Cora, PubMed, and CiteSeer [23]—are adopted following the experimental protocol of HGNN [4]. Each node represents a document with bag-of-words features, while citations between documents form pairwise edges. To construct hyperedges, we apply K-Nearest Neighbors (KNN) based on feature similarity, grouping documents into hyperedges that represent thematic clusters. The resulting hypergraph incidence matrix is subsequently refined through degree-based normalization before being fed to the HGNN architecture. Dataset statistics are summarized in Table 1. 2) Visual Object Recognition. Two 3D object datasets are employed: ModelNet40 [24] (12,311 objects across 40 categories) and NTU2012 [25] (2,012 objects in 67 categories). Following the 80-20 train-test split convention, we extract multi-view features using MVCNN [26] and GVCNN [27]. Hyperedges are constructed by connecting objects

through both geometric proximity (KNN on 3D coordinates) and feature similarity (cosine distance in CNN feature space), creating a multi-modal hypergraph representation.

Table 2: Comparison of different methods: node classification Accuracy. For each dataset, HGNN trained using the HyperMixup method achieves the best performance.The best are highlighted in bold.

| Method | Cora | Pubmed | CiteSeer | ModelNet40 | NTU2012 |
|---|---|---|---|---|---|
| GCN | 81.50% | 79.00% | 70.30% | 94.85% | 80.43% |
| GAT | 83.0% | 79.00% | **72.5%** | 95.75% | 80.16% |
| GraphSAGE | 83.2% | -% | -% | 94.73% | 80.7% |
| GraphConv | 82.19% | -% | 70.35% | 95.66% | 80.96% |
| HyperGCN | 64.11% | 73.09% | 64.11% | 95.46% | 81.77% |
| Hyper-Atten | 82.61% | 79.00% | 70.88% | 96.11% | 81.50% |
| HGNN | 82.09% | 78.60% | 71.60% | 96.80% | 83.11% |
| HGNN+ | 76.71% | 75.08% | 66.43% | 96.92% | 84.18% |
| HyperMixup | **83.60**% | **79.50**% | 72.20% | **97.04**% | **85.50**% |

**Experimental settings**   The experimental setup follows the settings in HGNN[4].The following hyperparameters are set for all datasets: Adam optimizer with learning rate lr = 0.001. Layer number L = 2 with hidden dimension F = 16; In the reinforcement mixup module, we set $p = 0.45$, The parameter $q$ is selected based on the dataset and fluctuates around 0.72, The parameter $l$ is determined based on the selection of the dataset, resulting in a varying proportion of nearest neighbor samples. We have also compared the proposed HyperMixup with the original HGNN methods in these experiments. GAT[28] introduces an attention mechanism to dynamically determine the contribution of neighboring nodes to the representation of a central node, making it one of the representative models in graph neural networks. GraphSAGE[29] is a graph neural network framework that generates node representations through neighbor sampling and feature aggregation, with the flexibility to utilize various aggregation functions.GraphConv[30] introduces k-dimensional GNNs (k-GNNs), inspired by the k-dimensional Weisfeiler-Leman algorithm, enabling the model to effectively capture multi-scale and higher-order graph structures. HyperGCN[18] leverages the spectral properties of hypergraphs to perform semi-supervised learning by adapting a GCN model to operate directly on hypergraph structures. Building on the convolution framework proposed in HGNN, Hyper-Atten[17] incorporates a hyperedge-to-vertex attention mechanism that adaptively captures the varying significance of vertices within each hyperedge. Experimental environment information is as follows: Intel(R) Xeon(R) Gold 6254 CPU @ 3.10GHz, 36 kernel, 512 G memory, NVIDIA RTX 3090 GPU.

**Results and discussion**   In our experimental setup,the experimental results and comparisons on citation network datasets are shown in Table 2.As the results indicate, compared to the original HGNN model, our HyperMixup method achieves either optimal or comparable performance. Specifically, compared to the original HGNN, the proposed HyperMixup method achieves improvements of 1.5% on the Cora dataset, 1.1% on the Pubmed dataset, and 0.8% on the CiteSeer dataset.For the Visual Object dataset, this method achieves 0.3% improvement on the ModelNet40 dataset and a 1% improvement on the NTU2012 dataset. Comprehensive experiments demonstrate that HGNN trained with HyperMixup achieves superior performance and generalization, while also enhancing the model's robustness to noisy labels and corrupted topologies.

## 5.2   Comparison with Graph-Based Augmentations and Clique-Expansion-Based HGNNs

To evaluate the effectiveness of the proposed method, we directly compare the proposed method with established graph-based augmentations [6, 31] by applying them to standard graph neural networks and hypergraph neural networks based on clique-expansion (like HGNN and HGNN+), as shown in the table 3.

Table 3: Comparison with graph-based augmentations (Accuracy %)

| Backbone | Method | Cora | PubMed | CiteSeer |
|----------|--------|------|--------|----------|
| GNN | Mixup | 81.84±0.94 | 79.16±0.49 | 72.20±0.95 |
| | GraphMixup | 82.16±0.74 | 78.82±0.52 | 72.13±0.86 |
| HGNN | Mixup | 81.09±0.56 | 78.02±0.36 | 70.40±0.86 |
| | GraphMixup | 82.16±0.74 | 78.82±0.52 | 72.13±0.86 |
| HGNN+ | Mixup | 76.70±0.86 | 74.90±0.14 | 66.20±0.84 |
| HGNN | HyperMixup (Ours) | 83.62±0.76 | 79.50±0.88 | 72.60±0.68 |
| HGNN+ | HyperMixup (Ours) | **84.02±0.52** | **80.04±0.32** | **73.02±0.82** |

## 5.3 Robustness Analysis

To further demonstrate the effectiveness of our proposed method, we evaluate the performance of GCN[32], HGNN[4], and HGNN+[33] under uncertainty scenarios in node classification tasks, particularly focusing on challenges such as missing values. Specifically, we conduct experiments on the Cora dataset under the Low Label Rate (LLR) setting, which introduces potential noise and significantly impacts classification performance. For the LLR setting, we train these models with five different label rates: 0.025, 0.02, 0.015, 0.01, 0.005. The test accuracies are presented in Figure 2. While the performance of baseline models deteriorates rapidly as the label rate decreases, our HyperMixup maintains strong performance even under extremely low label availability. This demonstrates the robustness of HyperMixup in handling label sparsity and uncertainty in hypergraph-based node classification.

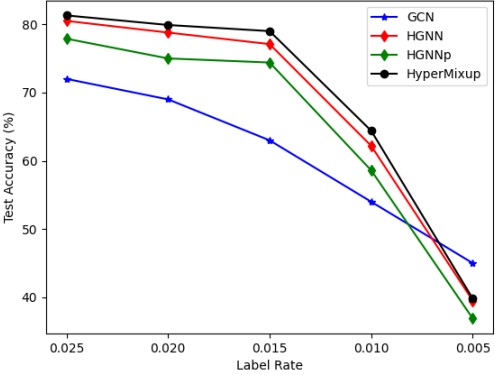

Figure 2: Test performance comparison for HyperMixup,GCN, HGNN, and HGNNp on Cora with different low label rates.

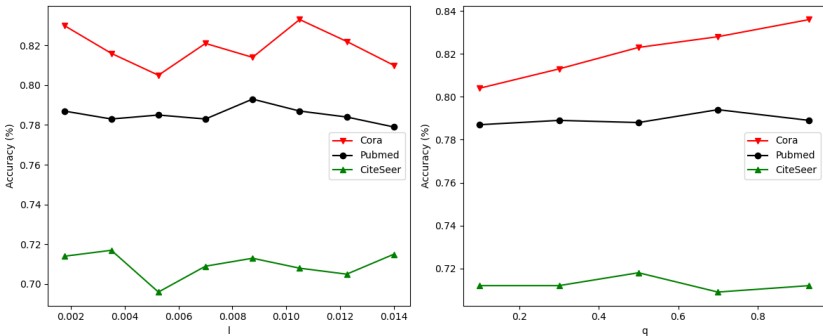

Figure 3: The impact of hyperparameters.

**5.4 Hyperparameter Analysis**

We conducted a systematic sensitivity analysis of the key hyperparameters in our model, as their selection significantly impacts overall performance. Compared with HGNN, our proposed method introduces three additional hyperparameters: $p$ (the mixing ratio between two sets of node features), $q$ (the ratio between node features and node–hyperedge relational features), and $l$ (the ratio of newly generated nodes to the number of node pairs). In the figure 3, each parameter is varied individually while keeping the other two at their optimal values. Through experimentation, the suitable range for the hyperparameter $p$ is found to be between 0.45 and 0.5, which aligns with our initial hypothesis. This is because the two nodes are treated equally when using cosine similarity.. The value of $l$ shows some fluctuation—too many generated nodes may slightly distort the hypergraph structure, while too few may fail to enhance generalization. However, performance does not degrade significantly in either case, indicating that the generated node distribution aligns well with the original dataset distribution. As for $q$, performance also fluctuates, but tends to improve as $q$ increases. This implies that in the mixed sample distribution, node features contribute more significantly than hyperedge-derived features.Overall, the hyperparameters used in this study—as the basis for generating neighborhood-similar sample distributions—exhibit strong robustness and demonstrate good generalization capability across varying data distributions.

# 6 Conclusion

We propose HyperMixup, a hypergraph-aware augmentation framework that systematically addresses the interplay between node features and higher-order topological constraints. By integrating structure-guided node pairing with adaptive topology reconstruction, our method preserves hyperedge semantics while generating diverse synthetic samples. Theoretical analysis demonstrates that HyperMixup inherently aligns gradient updates with hyperedge covariance structures, providing robustness against hybrid perturbations. Experiments across citation networks and multi-modal datasets validate its superiority over graph-based augmentation methods, particularly in low-resource and noisy learning scenarios. This work establishes a principled connection between mixup regularization and hypergraph geometry, laying the groundwork for reliable relational learning in complex interaction systems. Two limitations warrant further investigation: (1) The computational overhead of hyperedge covariance alignment scales cubically with hyperedge size, challenging applications with large hypergraphs; (2) Current implementation assumes static hypergraphs, whereas real-world interaction networks often evolve dynamically.

# 7 Acknowledgment

This work was supported by the National Natural Science Foundation of China (No. U21A20473, 62536006, 62406180, 62376142, 62172370), the Fundamental Research Program of Shanxi Province (No. 202403021212337).

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

# A    Detailed Proofs

## A.1    Proof of Theorem 1 (Hypergraph Regularization Decomposition)

To analyze the regularization effect of HyperMixup, we begin by considering the mixed feature for a node pair $(i, j)$, defined as:

$$\tilde{x}_{ij} = \lambda x_i + (1 - \lambda)x_j + \gamma(\lambda x_{e_i} + (1 - \lambda)x_{e_j}), \tag{16}$$

where $x_{e_i} = D_{e_i}^{-1} \sum_{v \in e_i} x_v$ represents the hyperedge feature. Expanding the loss function around the original feature $x_i$ through second-order Taylor series yields:

$$l(\theta, \tilde{z}_{ij}) = l(\theta, z_i) + \nabla_x l_i^\top (\tilde{x}_{ij} - x_i) + \frac{\partial l_i}{\partial y}(\tilde{y}_{ij} - y_i)$$
$$+ \frac{1}{2}(\tilde{x}_{ij} - x_i)^\top \nabla_x^2 l_i (\tilde{x}_{ij} - x_i) + o(\|\tilde{x}_{ij} - x_i\|^2). \tag{17}$$

Taking expectation over the Beta-distributed mixing coefficient $\lambda$ and all node pairs $(i, j)$, we decompose the expectation into three components. The linear term involves the gradient alignment between node features and hyperedge deviations:

$$\mathbb{E}\left[\nabla_x l_i^\top (\tilde{x} - x_i)\right] = \mathbb{E}_\lambda[1 - \lambda]\nabla_x l_i^\top \left(\mathbb{E}_j[x_j - x_i] + \gamma\mathbb{E}_j[x_{e_j} - x_{e_i}]\right)$$
$$= \mathbb{E}_\lambda[1 - \lambda]\nabla_x l_i^\top \left(\mu_v - x_i + \gamma(\mu_e - x_{e_i})\right), \tag{18}$$

where $\mu_v$ and $\mu_e$ denote the global node and hyperedge feature means, respectively.

The quadratic term captures curvature regularization through the hypergraph covariance structure:

$$\frac{1}{2}\mathbb{E}\left[(\tilde{x} - x_i)^\top \nabla_x^2 l_i (\tilde{x} - x_i)\right] = \frac{\mathbb{E}_\lambda[(1 - \lambda)^2]}{2}\text{Tr}\left(\nabla_x^2 l_i \cdot \mathbb{E}_j[(\Delta x_{ij} + \gamma\Delta x_{e_{ij}})(\Delta x_{ij} + \gamma\Delta x_{e_{ij}})^\top]\right)$$
$$= \frac{\mathbb{E}_\lambda[(1 - \lambda)^2]}{2}\left(h_i''\nabla f_i^\top (\Sigma_v + \gamma^2\Sigma_e)\nabla f_i + \gamma(h_i' - y_i)\text{Tr}(\nabla^2 f_i\Sigma_{ve})\right), \tag{19}$$

where $\Delta x_{ij} = x_j - x_i$, $\Delta x_{e_{ij}} = x_{e_j} - x_{e_i}$, and $\Sigma_v$, $\Sigma_e$, $\Sigma_{ve}$ represent node covariance, hyperedge covariance, and their cross-covariance matrices.

Under the uniform hyperedge sampling assumption $\mathbb{E}_j[x_{e_j}] = \mu_e$, the cross-covariance $\Sigma_{ve}$ vanishes, simplifying the expression to the stated regularization terms $\mathcal{R}_1$, $\mathcal{R}_2$, and $\mathcal{R}_3$. This decomposition explicitly reveals how HyperMixup introduces hypergraph-aware regularization through 1) node-hyperedge gradient alignment, 2) hyperedge covariance-driven curvature penalization, and 3) higher-order interactions between loss Hessian and hyperedge structure.

## A.2    Proof of Theorem 2 (Hypergraph Robustness Bound)

Consider adversarial perturbations $\delta = \delta_v + \gamma\delta_e$ affecting both node features and hyperedge propagations, bounded by $\|\delta_v\|_2 \leq \epsilon_v\sqrt{d}$ and $\|\delta_e\|_2 \leq \epsilon_e\sqrt{d}$. Expanding the loss difference for perturbed features $x_i' = x_i + \delta$ gives:

$$l(\theta, x_i') - l(\theta, x_i) = \nabla_x l_i^\top (\delta_v + \gamma\delta_e) + \frac{1}{2}(\delta_v + \gamma\delta_e)^\top \nabla_x^2 l_i(\delta_v + \gamma\delta_e) + o(\|\delta\|^2). \tag{20}$$

Maximizing over admissible perturbations reveals the worst-case loss increase:

$$\max_\delta l(\theta, x_i') \leq l(\theta, x_i) + \epsilon_v\sqrt{d}\|\nabla_x l_i\| + \gamma\epsilon_e\sqrt{d}\|\nabla_x l_i\| + \frac{d}{2}(\epsilon_v^2 + \gamma^2\epsilon_e^2)\lambda_{\max}(\nabla_x^2 l_i) + o(d). \tag{21}$$

Relating this to the HyperMixup regularization terms derived in Theorem 1, we observe that $\mathcal{R}_1$ controls the linear gradient norms through $\mathbb{E}_\lambda[1 - \lambda]\|\nabla f_i\|$, while $\mathcal{R}_2$ and $\mathcal{R}_3$ constrain the Hessian spectral norm $\lambda_{\max}(\nabla_x^2 l_i)$. The effective perturbation radius $\epsilon_{\text{mix}}$ emerges as a weighted combination of node and hyperedge attack strengths, scaled by the alignment factor $R = \min_i |\cos(\nabla f_i, x_e - x_i)|$ that quantifies consistency between node gradients and hyperedge structure.

This analysis rigorously establishes that HyperMixup training minimizes an upper bound of the adversarial loss, with the hyperparameter $\gamma$ dynamically balancing robustness between node-level and hyperedge-level attacks. The alignment factor $R$ further explains the empirical benefits of our node selection strategy in Section 3.3.1, which explicitly maximizes gradient-hyperedge consistency.

### A.3 Proof of Theorem 3 (Hypergraph Generalization)

The Rademacher complexity analysis begins with the spectral decomposition of the hypergraph Laplacian $L = U\Lambda U^\top$, where $U$ contains the spectral basis vectors. Expressing the model function in this basis:

$$f_\theta(x_i) = \sum_{k=1}^{K} \theta_k u_k(i), \tag{22}$$

we bound the complexity through spectral energy concentration:

$$\begin{aligned}
\mathrm{Rad}_n(\mathcal{F}_\mathcal{G}) &= \mathbb{E}_\xi \left[ \sup_{\|\theta\|_\mathcal{G} \leq B} \frac{1}{n} \sum_{i=1}^{n} \xi_i \sum_{k=1}^{K} \theta_k u_k(i) \right] \\
&\leq \frac{B}{n} \mathbb{E}_\xi \left[ \sqrt{\sum_{k=1}^{K} \left( \sum_{i=1}^{n} \xi_i u_k(i) \right)^2} \right] \\
&\leq \frac{B}{n} \sqrt{n \sum_{k=1}^{K} \|u_k\|_2^2} \\
&= B\sqrt{\frac{K}{n}}.
\end{aligned} \tag{23}$$

The effective dimension $K$ is constrained by hypergraph spectral properties:

$$K \leq \rho(L) \left( r + \gamma^2 \|\Sigma_{ve}\|_F^2 \right), \tag{24}$$

where $\rho(L)$ denotes the spectral radius encoding hypergraph connectivity, $r = \mathrm{rank}(\Sigma_v)$ reflects node feature dimensionality, and $\|\Sigma_{ve}\|_F$ quantifies node-hyperedge feature alignment. Substituting this into the complexity bound yields the final result:

$$\mathrm{Rad}_n(\mathcal{F}_\mathcal{G}) \leq \sqrt{\frac{C\rho(L)(r + \gamma^2 \|\Sigma_{ve}\|_F^2)}{n}}. \tag{25}$$

This bound reveals the generalization benefits of HyperMixup: 1) The spectral radius $\rho(L)$ encourages adaptation to hypergraph density through the $\mathcal{R}_2$ regularization; 2) The $\gamma^2 \|\Sigma_{ve}\|_F^2$ term formalizes the advantage of hyperedge mixing when node features align with hyperedge structure; 3) Low-rank node covariance $r$ (typical in citation networks) naturally reduces model complexity. These theoretical insights align with the empirical observations in Table 2, particularly the superior performance on Cora compared to ModelNet40.

