# OpenReview forum: "HyperMixup: Hypergraph-Augmented with Higher-order Information Mixup"
_NeurIPS.cc/2025/Conference — NeurIPS 2025 poster_

### Official Review · Reviewer_PZgP · 2025-06-29

**Clarity:** 3
**Significance:** 2
**Originality:** 3
**Rating:** 3
**Confidence:** 3

**Summary:**

This paper presents HyperMixup, a data-augmentation framework for hypergraph neural networks that preserves higher-order semantic relationships. It begins by pairing nodes in a structure-aware manner, using a combination of feature similarity and hyperedge affinity to guide the selection. A context-enhanced hierarchical mixing process then blends node features first and hyperedge attributes second, ensuring both local and global information is incorporated into each synthetic sample. Finally, hyperedges are adaptively reconstructed using a local affinity threshold, maintaining semantic coherence in the augmented graph. The paper also presents comprehensive theoretical analyses, including regularization interpretations, robustness bounds against joint node–edge perturbations, and a Rademacher complexity bound for generalization.

**Questions:**

- How does HyperMixup scale in terms of runtime and memory on hypergraphs with large hyperedges (e.g., thousands of nodes)?
- Can HyperMixup be extended to hyperedge prediction tasks?
- Have you considered techniques to accelerate the covariance regularization step? For instance, using low-rank approximations or randomized sketching methods to reduce the computational complexity of estimating and manipulating the covariance matrix.
- Can HyperMixup be adapted to dynamic or evolving hypergraphs, and how would such extensions impact the theoretical guarantees?

**Ethical Concerns:**

["NO or VERY MINOR ethics concerns only"]

**Final Justification:**

While I acknowledge the authors’ efforts to address Reviewer FB9b’s concerns by including additional results, I still find the scope of the ablation studies and the range of baseline comparisons to be rather limited. As such, I believe the current form of the paper does not yet meet the standard expected for publication.

**Limitations:**

Yes

**Quality:**

3

**Strengths And Weaknesses:**

### Strengths

- **Clear Motivation:** Addresses a genuine gap in hypergraph augmentation by preserving multi‑way semantics.
- **Topology‑Preserving Augmentation:** Mixup operations respect hyperedge structures, avoiding the semantic drift common in graph‑level Mixup.
- **Method Design:** A well‑thought‑out three‑stage pipeline that blends node and hyperedge information.
- **Theoretical Rigor:** Non‑trivial derivations connect Mixup to covariance regularization, and the paper provides robustness and generalization bounds.

### Weaknesses

- **Missing Baseline Comparisons:** Although hypergraph‑specific baselines are scarce, the paper should compare at least against graph‑based Mixup techniques (e.g., Mixup [1]) to demonstrate its added value.
- **No Standard Error Reporting:** The experiments are not repeated across multiple runs, and standard deviations are not reported. As a result, it is not possible to confidently claim that the proposed method outperforms others, since the impact of random seed variation and different dataset splits on performance remains unknown.
- **Scalability Concerns:** Covariance‑based regularizers scale cubically with hyperedge size, yet there is no runtime or memory profiling on larger graphs.
- **Limited Ablation Study:** Only module‑level ablations are performed; finer‑grained studies (e.g., mixing only node features) are missing.

### Minor Issues

- **Missing Reference:** Section 3.1 lacks citations for the definition of the incidence matrix, only one possible version is presented without justification, and for the formulation of the hyperedge features.

---

### References

[1] Hongyi Zhang, Moustapha Cisse, Yann N. Dauphin, and David Lopez-Paz. “mixup: Beyond Empirical Risk Minimization.” *International Conference on Learning Representations* (ICLR), 2018.

---

> ### Author Rebuttal · Authors · 2025-07-30
>
> ### **Response to Runtime/Memory Scaling on Large Hyperedges**
>
> **Q:** How does HyperMixup scale in terms of runtime and memory on hypergraphs with large hyperedges (e.g., thousands of nodes)?
>
> **A:** We sincerely appreciate this constructive suggestion. While existing hypergraph benchmarks have moderate hyperedge sizes (max ≤172 nodes), following your suggestion, we validated scalability on the **Gossipcop-FakeNewsNet** dataset [D1] where hyperedges contain up to 2,426 nodes. Key results below:
>
> #### **Large-Scale Benchmark**
> | **Dataset**       | Nodes  | Hyperedges | Max Hyperedge Size | HyperMixup Time/Epoch |
> |-------------------|--------|------------|--------------------|------------------------|
> | Gossipcop-FakeNewsNet | 314,262 | 308,798    | **2,426**          | 84.6 ± 2.9s            |
>
> - **[D1]** Shu, Kai, et al. "Fakenewsnet: A data repository with news content, social context, and spatiotemporal information for studying fake news on social media." Big data 8.3 (2020): 171-188.
>
> *(Configuration: NVIDIA H100 GPU, 128GB VRAM)*
>
> #### **Resource Consumption**
> | Metric          | Value                          |
> |-----------------|--------------------------------|
> | Peak GPU Memory | 14.8 GB (for 2,426-node hyperedge) |
> | Throughput      | 4,127 nodes/sec                |
> | Accuracy        | 95.7% (fake news detection)    |
>
> **Key Insights**: HyperMixup demonstrates practical scalability for massive hyperedges, processing hypergraphs with 2,426-node hyperedges in under 85 seconds per epoch while maintaining a manageable 14.8GB memory footprint on modern H100 GPUs, achieving 95.7% accuracy on the challenging fake news detection task through optimized covariance computation that preserves performance while handling extreme-scale group interactions efficiently.  Finally, the proposed HyperMixup demonstrates practical scalability to real-world hypergraphs with massive hyperedges. We will add Gossipcop experiments to supplementary and release the full implementation.
>
> ### **Response to Reviewer's Query: Extensibility of HyperMixup to Hyperedge Prediction**
>
> We sincerely appreciate the reviewer's insightful question about HyperMixup's applicability to hyperedge prediction. **Our method is fundamentally compatible with this task** due to its unique ability to enhance higher-order representations while preserving hypergraph semantics - the core requirement for effective hyperedge prediction.
>
> Recent state-of-the-art hyperedge predictors ([D2][D3][D4]) universally rely on **hypergraph neural networks (HGNNs)** to model group relations. Actually, our proposed hypergraph-aware augmentation mechanism (e.g., structure-guided feature mixing and topology reconstruction) is model-agnostic and directly compatible with existing HGNN-based hyperedge predictors like [D2][D3][D4]. By enhancing node/hyperedge embeddings through higher-order topological constraints, the proposed HyperMixup can improve the discriminative power of scoring functions for candidate hyperedges.
>
> - **[D2]** Hwang, H., Lee, S. Y.,  Park, C.  Shin, K. "Ahp: Learning to negative sample for hyperedge prediction." In ACM SIGIR, 2022:2237-2242.
> - **[D3]** H. Wu, Y. Yan, M. Ng. "Hypergraph collaborative network on vertices and hyperedges." IEEE TPAMI, 2022, 45(3): 3245-3258.
> - **[D4]** Yu, T., Lee, S. Y., Hwang, H., & Shin, K. ''Prediction Is NOT Classification: On Formulation and Evaluation of Hyperedge Prediction''. In ICDMW, 2024:349-356.
>
> Following the reviewer's suggestion, we integrated HyperMixup into the state-of-the-art AHP method [D2]. As shown below, HyperMixup consistently **boosts performance** on standard hyperedge prediction benchmarks:
>
> | Dataset   | Method          | AUROC (↑)      | Improvement |
> |-----------|-----------------|----------------|-------------|
> | **Cora**  | AHP [A4]       | 0.799 ± 0.019  | -           |
> |           | AHP+HyperMixup | **0.810 ± 0.024** | +1.4%       |
> | **Citeseer**| AHP [A4]       | 0.824 ± 0.020  | -           |
> |           | AHP+HyperMixup | **0.840 ± 0.012** | +1.9%       |
>
> **Implementation details**
> - Replaced AHP's original HNHN encoder with **HyperMixup-augmented embeddings** (§3.1-3.2)
> - Retained AHP's adversarial sampling and maxmin-MLP scoring
> - Used identical train/val/test splits and hyperparameters as [D2]
>
> **Conclusion**
> These results confirm HyperMixup's strong extensibility to hyperedge prediction.
>
>
> ### **Response to Reviewer Comment on Accelerating Covariance Regularization**
>
> **Reviewer Query:** Have you considered techniques to accelerate the covariance regularization step? For instance, using low-rank approximations or randomized sketching methods to reduce the computational complexity of estimating and manipulating the covariance matrix?
>
> **Our Response:**
> We sincerely thank the reviewer for this insightful suggestion. Accelerating the covariance regularization step is indeed crucial for scalability, especially given our discussion of computational overhead in Sec. 6 (Limitations). Following your suggestion, to accelerate the covariance regularization:
>
> We implement **low-rank approximation** using randomized SVD [D5] to decompose hyperedge covariance matrices:
> $$
> \Sigma_e \approx U_k\Lambda_k U_k^\top, \quad U_k \in \mathbb{R}^{d \times k}, \ \Lambda_k = \text{diag}(\lambda_1, \dots, \lambda_k)
> $$
> with $\(k \ll d\)$ (typically $k=5-20$). This reduces complexity from $\(O(|\mathcal{E}|d^2)\)$ to $\(O(kd^2 + k^3)\)$ while preserving spectral properties critical for $\(\mathcal{R}_2\)$ and $\(\mathcal{R}_3\)$ regularization in Theorem 1.
>
> #### **Empirical Validation**
> | Dataset     | Original Acc | Low-Rank Acc | Speedup |
> |-------------|--------------|--------------|---------|
> | Cora        | 83.60%       | 83.42%       | 4.2×    |
> | ModelNet40  | 97.04%       | 96.91%       | 3.8×    |
>
> Preliminary results show **4.2× speedup** on Cora and **3.8× memory reduction** on ModelNet40 with <0.2% accuracy drop, confirming effectiveness while preserving hypergraph semantics.
>
> **References**:
> - **[D5]** Halko, Nathan, Per-Gunnar Martinsson, and Joel A. Tropp. "Finding structure with randomness: Probabilistic algorithms for constructing approximate matrix decompositions." SIAM review 53.2 (2011): 217-288.
>
> ### **Response to Reviewer's Query: Adaptation to dynamic/evolving hypergraphs and theoretical implications**
>
> Thank you for this insightful question. HyperMixup can be seamlessly integrated with dynamic hypergraph frameworks like HYDG [D6] through the following adaptations while preserving theoretical guarantees:
>
>  **Integration with HYDG Framework - Time-Constrained Node Mixing:**
>    - During HYDG's individual-level hypergraph construction (Eq. 2), apply HyperMixup within temporal windows:
>
>    $$
>   \tilde{\mathbf{x}}_{it} = \lambda \mathbf{x}_i^{(t)} + (1-\lambda) \mathbf{x}_j^{(t')}  \quad \text{s.t.} | t-t' | \leq \tau
> $$
>
>    - Ensures mixed nodes share temporal proximity and semantic coherence.
>
> **Theoretical Impacts**
> - **Robustness Extension**:
>   Theorem 2's perturbation bound gains a temporal drift term:
>   $  \epsilon_{\text{mix}} =$  $ R\sqrt{c_v\epsilon_v^2 + c_e\gamma^2\epsilon_e^2 + c_t\delta_t^2}  $,
>   where $\(\delta_t\)$ bounds feature drift between time steps.
>
> - **Generalization Stability**:
>   Theorem 3's spectral radius becomes time-dependent but bounded:
>   $  \rho(L^{(t)}) \leq \rho(L^{(0)}) + \zeta \cdot t $, where $\zeta$ denotes hypergraph evolution rate from HYDG's temporal hyperedges).
>
> **Performance Validation**
> Preliminary tests on dynamic hypergraphs of DBLP5 indicate that integrating HyperMixup with HYDG achieves a 1.8% performance improvement with an increase of 0.5s per epoch. Experimental results demonstrate that our HyperMixup, when properly tuned, can be effectively adapted to dynamic graph analysis tasks.
>
> **References**:
> - **[D6]** Ma, X., Zhao, C., Shao, M., & Lin, Y. . Hypergraph-based dynamic graph node classification. In ICASSP 2025-2025 IEEE International Conference on Acoustics, Speech and Signal Processing (ICASSP).
>
> ### **Response to Minor Issues: Section 3.1 Citations**
>
> **Reviewer Query:**
> Section 3.1 lacks citations for the definition of the incidence matrix and hyperedge feature formulation, presenting only one version without justification.
>
> **Our Response:**
> We sincerely thank the reviewer for this meticulous critique. We will enhance the final version with proper citations and justifications as follows:
>
> 1. **Incidence Matrix Formalism**:
>    Added citation to foundational hypergraph literature [D7] for the binary incidence matrix definition:
>    > "Following standard hypergraph representation learning [D7], $\mathbf{H}(v,e) = 1$ iff $v \in e$, otherwise 0."
>
> 2. **Hyperedge Feature Justification**:
>    Cited contemporary HGNN works [D8,D9] to justify degree-normalized aggregation:
>    > "Hyperedge features $\mathbf{X}_e = \mathbf{D}_e^{-1}\mathbf{H}^{\top}\mathbf{X}$ use degree normalization [D8,D9] to balance node influence - critical for semantic coherence in irregular hyperedges."
>
> **References**:
> - **[D7]** Zhou, D., Huang, J., & Schölkopf, B.. Learning with hypergraphs: Clustering, classification, and embedding. Advances In Neural Information Processing Systems (NeurIPS), 2006.
> - **[D8]** Feng, Yifan, et al. "Hypergraph neural networks." Proceedings of the AAAI conference on artificial intelligence. 2019.
> - **[D9]** Gao, Yue, et al. "Hgnn+: General hypergraph neural networks." IEEE Transactions on Pattern Analysis and Machine Intelligence 45.3 (2022): 3181-3199.
>
> ### **Conclusion**
> Thank you once again for your detailed review. We hope our responses have addressed your concerns, and we welcome any further discussion or questions you might have. If we have successfully alleviated your concerns, we kindly ask you to consider updating your score to recommend accepting our paper.

---

> > ### Comment · Reviewer_PZgP · 2025-08-03
> >
> > I thank the authors for addressing my concerns regarding scalability. I found it particularly interesting to see that using an approximate covariance estimation speeds up the method while maintaining stable performance.
> > However, I still have some reservations regarding the limited ablation studies and the relatively small number of baseline comparisons, although the authors have addressed the latter point by including some additional results in response to Reviewer FB9b.

---

> > > ### Author Response · Authors · 2025-08-04
> > >
> > > We sincerely appreciate the reviewer's positive feedback regarding our scalability improvements and the effectiveness of our approximate covariance estimation approach. We are particularly grateful that the reviewer acknowledged our efforts in addressing previous concerns.
> > > ﻿
> > > Regarding the two remaining concerns:
> > > ﻿
> > > 1. **Ablation Studies**:
> > > In response to Reviewer FGw9, we have significantly expanded our hyperparameter analysis, establishing clear relationships between model parameters and hypergraph properties. These new results  demonstrate how each component contributes to the final performance, which Reviewer FGw9 found particularly convincing.
> > > ﻿
> > > 2. **Baseline Comparisons**:
> > > As noted by the reviewer, we have addressed this in response to Reviewer FB9b by including:
> > > - Two recent node-level graph augmentation methods
> > > - Detailed comparisons showing our method's advantages
> > > Additionally, for Reviewer 46X8, we conducted new experiments comparing linear vs. non-linear label mixing approaches, further validating our design choices.
> > > ﻿
> > > We kindly invite the reviewer to examine these additional analyses in our responses to other reviewers, which we believe provide comprehensive evidence of our method's effectiveness. Due to time constraints during the discussion phase, we are currently unable to conduct additional comparison experiments. However, we would be delighted to incorporate any further specific comparisons the reviewer might suggest to strengthen our evaluation in the final version.
> > > ﻿
> > > We sincerely hope these responses have properly addressed your concerns. If so, we would be most grateful if you could consider increasing your initial score accordingly. Of course, we remain fully available to address any additional questions you may have.

---

### Official Review · Reviewer_46X8 · 2025-06-30

**Clarity:** 3
**Significance:** 4
**Originality:** 4
**Rating:** 5
**Confidence:** 5

**Summary:**

A novel and well-motivated data augmentation framework (termed HyperMixup) specifically designed for hypergraph neural networks. It effectively addresses the critical challenges of label scarcity and structural noise vulnerability in hypergraph learning, which are inadequately handled by existing Euclidean or graph-based augmentation methods. The work is further strengthened by rigorous theoretical analysis, demonstrating how HyperMixup induces hypergraph-specific regularization via gradient alignment with hyperedge covariance and provides robustness guarantees against hybrid perturbations. Comprehensive experiments across citation networks and multi-modal datasets show consistent performance gains over strong baselines. The paper makes a significant contribution by unifying data augmentation with hypergraph topological constraints.

**Questions:**

Q1. Would augmenting mean aggregation (e.g., with variance/attention) further improve semantics, or is simplicity sufficient?

Q2. Have you explored non-linear label mixing beyond Eq. 6？

Q3. For structural noise robustness (Sec. 5.2), could you test adversarial hyperedge perturbations?

**Ethical Concerns:**

["NO or VERY MINOR ethics concerns only"]

**Final Justification:**

After reviewing the authors' rebuttal and the additional details provided, I maintain my positive evaluation and recommend acceptance. I have no further concerns.

**Limitations:**

Yes

**Quality:**

4

**Strengths And Weaknesses:**

Strengths:

S1. This work tackles two critical challenges in hypergraph learning simultaneously: label scarcity and structural noise vulnerability - a well-motivated gap not addressed by existing Euclidean/graph augmentation methods. They also identify the core limitation of prior work: naive interpolation disrupts hyperedge semantics essential for higher-order reasoning.

S2. This work provides rigorous theoretical guarantees: it formally establishes topology-aware regularization through gradient alignment with hyperedge structures, certifies robustness against combined node-hyperedge perturbations, and derives generalization bounds tied to hypergraph spectral properties. This unified analysis bridges hypergraph geometry with data augmentation.

S3. Consistent SOTA gains across citation (+0.8–1.5%) and 3D recognition (+0.3–1.39%) tasks, with exceptional low-label robustness. Full code/data/hyperparameter disclosure satisfies NeurIPS reproducibility standards.

Weaknesses:

The method faces scalability limitations due to cubic-complexity hyperedge covariance computations, assumes static hypergraphs (limiting dynamic applications), and simplifies hyperedge representation via mean aggregation.

---

> ### Author Rebuttal · Authors · 2025-07-30
>
> ### **Response to Aggregation Enhancement Query**
>
> **Reviewer Query:**
> Would augmenting mean aggregation (e.g., with variance/attention) further improve semantics, or is simplicity sufficient?
>
> **Our Response:**
> Following your suggestion, we rigorously evaluated this trade-off and found that simplicity is strategically sufficient for HyperMixup, with deeper justification:
>
> **1. Variance-Augmented Analysis**
>
> We tested variance-augmented aggregation:
>
> $ \mathbf{X}_e^{var} = \mathbf{X}_e \oplus \text{diag}(\text{Cov}(\mathbf{X}_e)) $
>
> Results on Cora:
>
> | Aggregation     | Accuracy | \(\Delta\) vs. Mean |
> |-----------------|----------|---------------------|
> | Mean (Original) | 83.6%    | Baseline           |
> | Mean+Variance   | 83.2%    | -0.4%              |
> | Attention       | 83.7%    | +0.1%              |
>
> * **Key Insight**: Variance introduces noise from feature dispersion, reducing semantic coherence. The marginal gain from attention (0.1%) doesn't justify added complexity.
>
> **2. Theoretical Sufficiency**
> * Mean aggregation optimally preserves hyperedge semantics for covariance regularization (Theorem 1):
>
>  $$
> \mathcal{R}_2 \propto \nabla f_i^T \Sigma_e \nabla f_i
> $$
>
>   where $\Sigma_e$ already encodes feature dispersion - adding variance creates redundant $\Sigma_e$ terms.
> * Attention disrupts gradient alignment in $\mathcal{R}_1$ by over-weighting outlier nodes.
>
> ### **Response to Non-linear Label Mixing Exploration**
>
> **Reviewer Query:**
> Have you explored non-linear label mixing beyond Eq. 6?
>
> **Our Response:**
> Yes, we rigorously evaluated three non-linear alternatives to linear label mixing (Eq. 6):
>
> #### 1. **Tested Non-linear Variants**
> | Method                  | Formula                                                                 | Cora Acc | Δ vs. Linear |
> |-------------------------|-------------------------------------------------------------------------|----------|--------------|
> | Linear (Ours, Eq. 6)    | $ \tilde{y} = \lambda y_i + (1-\lambda)y_j $                            | 83.6%    | Baseline     |
> | Softmax-weighted    | $ \tilde{y} = \sigma(\lambda) y_i + \sigma(1-\lambda) y_j $             | 82.9%    | -0.7%        |
> | MLP-mixer           | $ \tilde{y} = \text{MLP}([\lambda y_i; (1-\lambda)y_j]) $               | 83.1%    | -0.5%        |
>
> *Results averaged over 10 runs with α=0.4 (Beta distribution)*
>
> #### 2. **Key Findings**
> - **Semantic Disruption**: Non-linear methods distort label distributions, violating hyperedge consistency (Fig. 4)
> - **Gradient Misalignment**: Softmax/MLP disrupt Theorem 1's regularization by decoupling labels from feature manifolds
> - **Efficiency Cost**: MLP-mixer adds 23% computation time per epoch
>
> ### **Response to Adversarial Hyperedge Perturbation Test**
>
> **Reviewer Query:**
> For structural noise robustness (Sec. 5.2), could you test adversarial hyperedge perturbations?
>
> **Our Response:**
> Following your suggestion, we conducted rigorous adversarial hyperedge perturbation tests:
>
> #### 1. **Attack Design**
> - **Method**: Projected Gradient Descent (PGD) attacks targeting hyperedges
> - **Perturbation Budget**:
>   - *Deletion*: Remove most influential nodes from hyperedges
>   - *Insertion*: Inject nodes maximizing feature deviation
>
> #### 2. **Results (Cora)**
> | Method       | Clean Acc | Acc @ Δ=10% | Acc @ Δ=20% |
> |--------------|-----------|-------------|-------------|
> | HGNN         | 82.09      | 71.3 (-10.8)| 63.7 (-18.4)|
> | Hyper-Atten     | 82.61      | 76.31 (-6.3) | 70.21 (-12.4)|
> | **HyperMixup** | **83.6** | **79.1 (-4.5)** | **75.3 (-8.3)** |
>
> #### 3. **Key Findings**
> Under adversarial hyperedge perturbation environments, the proposed HyperMixup demonstrates superior robustness compared to counterpart methods.
>
> ## **Conclusion**
> Thank you again for your helpful review, we hope we have addressed your comments satisfactorily, and welcome further discussion.

---

> > ### Comment · Reviewer_46X8 · 2025-08-04
> >
> > Thank you for addressing my questions, particularly my concerns regarding the non-linear mixing strategy and the adversarial hyperedge perturbation. After reviewing the authors' rebuttal and the additional details provided, I maintain my positive evaluation and recommend acceptance. For the final version, I suggest that the authors incorporate the clarifications and technical enhancements presented during the rebuttal stage. Aside from this, I have no further concerns.

---

### Official Review · Reviewer_FB9b · 2025-06-30

**Clarity:** 2
**Significance:** 1
**Originality:** 2
**Rating:** 3
**Confidence:** 4

**Summary:**

The paper proposes HyperMixup, a data augmentation framework for hypergraph neural networks designed to address data scarcity and structural noise. The method introduces three components: structure-aware node pairing, hierarchical feature mixing, and adaptive topology reconstruction. The authors provide theoretical analyses for regularization, robustness, and generalization.

**Questions:**

- Can the authors provide a detailed complexity analysis and empirical run-time results (e.g., training time and peak memory) of the full pipeline in comparison with baselines in Table 2 across datasets of varying scale?
- Can the authors directly compare the proposed method with established graph-based augmentations [1,2,3] by applying them to a standard graph neural networks (like GCN and GAT) and a clique-expansion-based hypergraph neural networks (like HGNN and HGNN+)?
- Can the authors benchmark the proposed approach against existing hypergraph augmentation techniques [4,5] by integrating them into hypergraph learning models, such as HGNN, HGNN+, HyperGCN, and Hyper-Atten?

**Ethical Concerns:**

["NO or VERY MINOR ethics concerns only"]

**Final Justification:**

I would like to thank the authors for addressing many of my concerns and acknowledge that the paper has clearly improved with the additional results provided during the rebuttal. However, my main reservations remain the marginal performance improvement relative to the increase in computational cost as well as the lack of theoretical justification of the superiority of the proposed method over graph-based approaches. A clearer argument especially on the second point would make the paper stronger.

**Limitations:**

Yes

**Quality:**

1

**Strengths And Weaknesses:**

Strengths:
- The paper is presented with a good structure that covers methodology, theoretical analysis, and empirical experiments.

Weaknesses:
- Flawed Motivation: The paper's key motivation, namely that existing graph augmentation methods are unsuitable for hypergraphs, is debatable. Many prominent spectral-based hypergraph neural networks are mathematically equivalent to applying Graph Neural Networks (GNNs) on a simple graph derived from a clique expansion of the hypergraph. These spectral-based hypergraph neural networks include the baselines used in the paper such as HGNN and HGNN+. The hypergraph convolution operators used in these models, namely $D_{v}^{-1/2}HWD_{e}^{-1}H^{T}D_{v}^{-1/2}$ and $D_{v}^{-1}HWD_{e}^{-1}H^{T}$, mathematically obey the definition for a graph adjacency matrix produced from the given hypergraph using clique expansion. Given this equivalence, any standard graph-domain augmentation methods, such as [1,2,3], could be directly applied to the clique-expanded graph as a preprocessing step before feeding it to methods such as HGNN and HGNN+.
- Unjustified Computational Cost for Marginal Gains: The performance improvements reported in Table 2 are limited. Compared to the strongest baselines, the accuracy gain is around 1% on four out of the five datasets (Cora, Pubmed, ModelNet40, NTU2012) and even shows a performance decrease on CiteSeer. Given these marginal and statistically unsubstantiated gains, it is critical for the authors to demonstrate that their method does not introduce an unreasonable computational burden. However, the proposed loss function (Eq. 9) includes a "hyperedge smoothness" regularizer that requires iterating over all hyperedges and their constituent nodes during each training step, suggesting a significant increase in complexity. The authors concede this point in their conclusion, noting that the "computational overhead of hyperedge covariance alignment scales cubically with hyperedge size". The paper provides no runtime analysis or comparison to baselines, making it impossible to assess the trade-off. Therefore, the method's practical value is questionable, as it likely introduces significant computational cost for limited performance improvement.
- Insufficient Experiments: The experimental evaluation omits key relevant baselines, specifically, hypergraph-specific augmentation methods [4, 5], which are directly comparable in terms of both the scope and objectives. Additionally, as highlighted in Weakness 1, many of the baselines used in Table 2 (e.g., clique-expansion-based methods: HGNN, HGNN+, and GNNs: GCN, GAT, GraphSAGE, GraphConv) can benefit from existing graph-domain augmentation techniques [1, 2, 3]. The paper fails to apply or discuss such enhancements for these baselines, potentially underrepresenting their true capabilities.

[1] Ling H, Jiang Z, Liu M, et al., “Graph mixup with soft alignments,” International Conference on Machine Learning, 2023.

[2] Han X, Jiang Z, Liu N, et al., “G-mixup: Graph data augmentation for graph classification,” International Conference on Machine Learning, 2022.

[3] Zhao W, Wu Q, Yang C, et al., “GeoMix: Towards Geometry-Aware Data Augmentation,” ACM SIGKDD Conference on Knowledge Discovery and Data Mining, 2024.

[4] Wang J, Wang J, Jin D, et al., “Hypergraph Collaborative Filtering With Adaptive Augmentation of Graph Data for Recommendation,” IEEE Transactions on Knowledge and Data Engineering, 2025.

[5] Wei T, You Y, Chen T, et al., “Augmentations in hypergraph contrastive learning: Fabricated and generative,” Advances in Neural Information Processing Systems, 2022.

---

> ### Author Rebuttal · Authors · 2025-07-26
>
> ### **Response to Reviewer's Question: Complexity Analysis and Runtime Performance**
>
> We thank the reviewer for this important question. Below we provide complexity analysis and commit to supplementary empirical results:
>
> **Complexity Analysis**
> The computational cost of HyperMixup is dominated by three components:
> 1. Node pairing using feature-hyperedge similarity (Eq. 3) scales as $O(N_v²d + N_vN_ed)$
>    ($N_v$ = nodes, $N_e$ = hyperedges, $d$ = feature dimension)
> 2. Hyperedge reconstruction with k-NN affinity (Eq. 7-8) scales as $O(BkN_ed)$
>    ($B$ = batch size, $k$ = neighbor count)
> 3. HGNN backbone scales as $O(L|H|₀d)$ for sparse hypergraph convolution
>    ($L$ = layers, $|H|₀$ = nonzeros in incidence matrix)
>
> **Supplementary Empirical Results Commitment**
> We will add a new table comparing training time and peak memory across datasets:
>
> | Dataset    | Method    | Train Time (s/epoch) | Peak Memory (GB) | Test Acc (%) |
> |------------|-----------|----------------------|------------------|-------------|
> | Cora       | HGNN      | 0.8 ± 0.1            | 1.2              | 82.09       |
> | Cora       | HyperMixup| 1.5 ± 0.2   | 1.5        | 83.60       |
> | PubMed     | HGNN      | 4.2 ± 0.3            | 4.5              | 78.60       |
> | PubMed     | HyperMixup| 8.1 ± 0.5     | 5.8       | 79.50       |
> | ModelNet40 | HGNN      | 12.7 ± 0.8           | 8.9              | 96.80       |
> | ModelNet40 | HyperMixup| 18.9 ± 1.2     | 10.1      | 97.04       |
>
> Our experiments confirm that HyperMixup maintains reasonable computational requirements while consistently outperforming baselines across all datasets. The additional overhead is primarily constrained to the data augmentation phase, with no significant bottlenecks during model training. Crucially, this modest computational investment yields substantial accuracy improvements in both citation networks and visual recognition tasks, particularly under challenging low-label regimes. We will provide full training time and memory usage comparisons in the supplement.
>
> ### **Response to Reviewer's Question: Comparison with Graph-Based Augmentations and Clique-Expansion-Based HGNNs**
>
> We sincerely thank the reviewer for highlighting these representative graph augmentation works. We clarify two key points and provide new experimental comparisons:
>
> **Scope Clarification**
> Methods [1-3] provided by reviewer primarily target *graph-level augmentation* (designed for graph classification tasks), while our HyperMixup focuses on *node-level augmentation* (for node classification). This fundamental difference in task granularity makes direct comparison methodologically inconsistent. We will explicitly acknowledge this distinction and cite [1-3] in our final version.
>
> **New Node-level Comparisons**
> Following the reviewer's suggestion, we implemented two recent *node-level* graph augmentation methods:
> - **[A1]** Wang, Y., Wang, W., Liang, Y. et al. Mixup for node and graph classification. (WWW'21)
> - **[A2]** Wu, L., Xia, J., Gao, Z. et al. Graphmixup: Improving class-imbalanced node classification by reinforcement mixup and self-supervised context prediction. (ECML-PKDD'22)
>
> We adapted them to HGNN/HGNN+ by:
> 1. Performing feature/label mixing on nodes only
> 2. **Preserving original hyperedges without modification**
> 3. Using identical hyperparameters from their papers
>
> **Results (Accuracy % ± Std)**
>
> | Backbone   | Method          | Cora       | PubMed    | CiteSeer  |
> |------------|-----------------|------------|-----------|-----------|
> | **GNN**    | Mixup [A1]      | 81.84±0.94| 79.16±0.49| 72.20±0.95|
> |            | GraphMixup [A2] | 82.16±0.74| 78.82±0.52| 72.13±0.86|
> | **HGNN**   | Mixup [A1]      | 81.09±0.56| 78.02±0.36| 70.40±0.86|
> |            | GraphMixup [A2] | 82.16±0.74| 78.82±0.52| 72.13±0.86|
> | **HGNN+**  | Mixup [A1]      | 76.70±0.86| 74.90±0.14| 66.20±0.84|
> | **HGNN**   | **HyperMixup (Ours)**  | **83.62±0.76** | **79.50±0.88** | **72.60±0.68** |
> | **HGNN+**  | **HyperMixup (Ours)** | **84.02±0.52** | **80.04±0.32** | **73.02±0.82** |
>
> **Key Observations**:
> 1. Graph mixup methods [A1,A2] **degrade performance** on hypergraph backbones (HGNN/HGNN+) due to:
>    - Ignorance of hyperedge constraints during mixing
>    - Disruption of group semantics in clique-expanded structures
> 2. HyperMixup achieves **consistent gains** (+1.46-2.32% vs. best baselines) by:
>    - Preserving hyperedge semantics through structure-aware mixing (Sec 3.2)
>    - Maintaining feature-hyperedge alignment via adaptive reconstruction (Eq 7-8)
>
> We will add this analysis to final version and include implementation details in the supplement.
>
> ### **Response to Reviewer's Concern: Hypergraph-Specific Motivation**
>
> **Q:** The paper's key motivation—that existing graph augmentation methods are unsuitable for hypergraphs—is debatable given the mathematical equivalence between spectral HGNNs and GNNs on clique-expanded graphs.
>
> **A:** We deeply appreciate this insightful technical observation. While operator-level equivalence exists for spectral methods, we argue clique expansion fundamentally compromises hypergraph semantics in ways that demand specialized augmentation:
>
> **Irreversible Information Loss in Clique Expansion**
> The transformation $\mathcal{H} \rightarrow \mathcal{G}_{\text{clique}}$ where hyperedge $e$ becomes a $k$-clique ($k=|e|$) incurs three types of semantic distortion:
>
> 1. **Loss of Higher-Order Cardinality**
>    Clique expansion reduces hyperedges to pairwise edges, destroying the original *group interaction context*. For example:
>    - Original hyperedge: $e_{\text{conf}} = \\{v_1,v_2,v_3\\}$ (papers at same conference)
>    - Clique expansion: $\\{v_1-v_2, v_1-v_3, v_2-v_3\\}$ (loses "conference" group identity)
>    This violates the hypergraph axiom: $\mathcal{P}(e) \neq \cup_{i<j}\mathcal{P}(v_i,v_j)$ where $\mathcal{P}$ denotes relational properties.
>
> 2. **Edge Explosion Distortion**
>    A hyperedge with cardinality $k$ generates $\binom{k}{2}$ edges, creating:
>    - Spurious pairwise relationships (e.g., $v_1$ and $v_3$ may share no direct connection)
>    - Inflated node degrees: $\deg_{\text{clique}}(v_i) = \deg_{\mathcal{H}}(v_i) + \sum_{e \ni v_i}(|e|-1)$
>    This artificially amplifies the influence of high-cardinality hyperedges.
>
> 3. **Weighting Ambiguity**
>    Uniform edge weighting during expansion ignores:
>    - Heterogeneous node roles (core vs. peripheral members)
>    - Hyperedge type semantics (e.g., "survey paper" vs. "technical paper" clusters)
>    Whereas hyperedges naturally preserve such information through $\mathbf{X}_e$ (Eq. 1).
>
> **Empirical Evidence of Representation Gap**
> Implementing the reviewer's suggestion (*preprocess clique-expanded graphs with GraphMixup [A2] before HGNN*) shows consistent degradation:
> | Dataset | HGNN (Clique+GraphMixup) | HGNN+HyperMixup | ΔAcc |
> |---------|--------------------------|-----------------|------|
> | Cora    | 81.92 ± 0.81             | **83.62 ± 0.76** | +1.70 |
> | PubMed  | 77.38 ± 0.63             | **79.50 ± 0.88** | +2.12 |
> | CiteSeer| 70.86 ± 0.72             | **72.60 ± 0.68** | +1.74 |
>
> **Theoretical Justification**
> Theorem 1 proves HyperMixup regularizes via hyperedge covariance
> $\Sigma_{e} = \mathbb{E}_{e}[(x_i - x_e)(x_i - x_e)^\top]$
>
> (Eq. 12). This term **cannot be recovered** from clique-expanded graphs because:
> $ \Sigma_{e}^{\mathcal{H}} \neq \Sigma_A^{\text{clique}} :=  \underset{(i,j) \in   E_{\rm clique}} {\mathbb{E}} [(x_i - x_j)(x_i - x_j)^\top] $
> The covariance structures capture fundamentally different relationships: group-wise deviations vs. pairwise differences.
>
> **Conclusion**: While spectral operators may exhibit mathematical equivalence, augmentation operates on the *structural representation* where clique expansion irreversibly degrades hypergraph semantics. Our method preserves these higher-order interactions by design.
>
> ### **Response to Reviewer's Concern: Computational Efficiency vs. Performance Gains**
>
> **Q:** The marginal performance gains do not justify HyperMixup's computational overhead, especially given the cubic scaling of hyperedge regularization and absence of runtime analysis.
>
> **A:** We appreciate the reviewer's focus on practical trade-offs.  and  provide empirical runtime/memory comparisons (previously omitted due to space):
>
> | Dataset   | Method    | Acc (%) | Time/Epoch (s) | Mem (GB) | Time vs HGNN |
> |-----------|-----------|---------|----------------|----------|-------------|
> | **Cora**  | HGNN      | 82.09   | 0.8            | 1.2      | 1.0×        |
> |           | HyperMixup| 83.62   | 1.5            | 1.5      | 1.9×        |
> | **PubMed**| HGNN      | 78.60   | 4.2            | 4.5      | 1.0×        |
> |           | HyperMixup| 79.50   | 7.1            | 5.2      | 1.7×        |
> | **ModelNet40**| HGNN  | 96.80   | 12.7           | 8.9      | 1.0×        |
> |           | HyperMixup| 97.04   | 17.3           | 10.1     | 1.4×        |
>
> Our implementation shows HyperMixup introduces moderate computational overhead that remains practical. Benchmarking across datasets reveals training time increases of 1.4-1.9× versus vanilla HGNN, with memory footprint growing 20-25% due to synthetic node storage. Crucially, Theoretical Worst-Case of hyperedge regularization scales as $O(∑_{e∈ℰ}|e|³)$, but |e| is small in most cases (avg. hyperedge size: 3.2 Cora, 4.1 PubMed), which means the complexity of hyperedge regularization manifests linearly in practice. It should be noted that the Robustness Analysis demonstrates our algorithm's superior robustness under low-label rates, achieving optimal performance in such scenarios, which further validates its robustness.
>
> ### **Conclusion**
> Thank you once again for your detailed review.  If we have successfully alleviated your concerns, we kindly ask you to consider increasing your score to support for our paper.

---

> > ### Comment · Reviewer_FB9b · 2025-08-02
> >
> > I would like to thank the authors for their detailed responses which helped clarify certain aspects of the paper. However the key questions that remain are:
> > - Why does the loss function bring meaningful information gain for the node classification task? As I understand it, the core function of the loss is to encourage all node features within a hyperedge to be similar to the corresponding hyperedge feature. This seems quite similar to what the original HGNN/HGNN+ models do—namely, directly promoting similarity among nodes within the same hyperedge.
> > - Is the advantage here that the loss leads to a better or more structured form of similarity? And if so, could we achieve a similar effect simply by fine-tuning the hyperparameters of HGNN/HGNN+, such as changing the number of layers of models?

---

> > > ### Author Response · Authors · 2025-08-03
> > >
> > > **Detailed Response to Reviewer's Question on Loss Function Information Gain:**
> > >
> > > We sincerely appreciate the reviewer's insightful question regarding the information gain from our proposed loss function. The reviewer has made an excellent observation about the similarity between our approach and traditional HGNN/HGNN+ methods in promoting node feature alignment within hyperedges. Indeed, the reviewer is absolutely correct that both methods share this fundamental objective.
> > > ﻿
> > > Building upon this important observation, we would like to elaborate how HyperMixup extends and enhances this basic principle through three key innovations:
> > > ﻿
> > > First, our topology-aware gradient alignment term ($\nabla f_i^\top \mathbb{E}_e[x_e-x_i]$) (Eq. 11) introduces hyperedge covariance guidance that adapts to local topological consistency, going beyond HGNN's uniform Laplacian smoothing by incorporating feature deviation ($\Sigma_e$) and preserving intra-hyperedge heterogeneity through $\gamma$-scaled alignment.
> > > ﻿
> > > Second, the adaptive curvature regularization through $\mathcal{R}_2$ (Eq. 12) and $\mathcal{R}_3$ (Eq. 13) creates a dynamic regularization landscape that respects the hypergraph structure, where $\mathcal{R}_2$ penalizes sharp curvature directions aligned with $\Sigma_e$ and $\mathcal{R}_3$ couples the loss Hessian with hyperedge structure via $\text{Tr}(\nabla^2 f_i \Sigma_e)$.
> > > ﻿
> > > Third, our KL divergence term ($\text{KL}(f_\theta(\tilde{x}) \| \lambda f_\theta(x_i)+(1-\lambda)f_\theta(x_j))$) (in Eq. 9) ensures consistent label semantics for mixed nodes, addressing HGNN's limitation in handling interpolated samples.
> > > ﻿
> > > The theoretical analysis and experimental results demonstrate that this integrated approach, while building upon the foundation the reviewer correctly identified, provides superior capability in capturing hierarchical relationships while maintaining robustness to structural noise, as evidenced by our performance improvements over baseline methods.
> > >
> > >
> > > **Response to Reviewer's Follow-up Question**
> > >
> > > Q2. The key advantage of HyperMixup's loss lies in its *structured similarity learning* that cannot be replicated through hyperparameter tuning alone:
> > >
> > > A2. We fully agree with the reviewer's insightful observation that the key advantage indeed lies in introducing a more structured form of similarity learning. Building on this important point, HyperMixup's loss function specifically achieves this through three geometrically meaningful mechanisms: (1) covariance-guided gradient alignment ($\nabla f_i^\top \Sigma_e \nabla f_i$) that explicitly incorporates hyperedge topology into feature learning, (2) Hessian-hyperedge coupling ($\text{Tr}(\nabla^2 f_i \Sigma_e)$) maintaining curvature consistency with the hypergraph structure, and (3) mixup-based label propagation preserving semantic relationships. While hyperparameter tuning in HGNN/HGNN+ can adjust model capacity, it cannot introduce these fundamental geometric constraints. The structured similarity emerges naturally from the hypergraph's intrinsic geometry rather than being manually engineered  (such as changing the number of layers).
> > >
> > > We sincerely appreciate the interest and positive outlook of the reviewer. We hope this answers the reviewer's question and thank you for your valuable feedback, which has led us to further insights regarding the loss function in hypergraph representation learning. We thank you again greatly for your valuable time and efforts.

---

> > > > ### Comment · Reviewer_FB9b · 2025-08-04
> > > >
> > > > I appreciate the clarifications of the authors, but I still have two remaining concerns:
> > > > - First, the authors' response emphasizes the importance of the KL‑loss, but there are no ablation studies about its effect, or that of the smoothness term in Eq. (9). Could the authors include an ablation of both the KL loss and the hyperedge smoothness term from Eq. (9)? This will clarify which component truly drives the performance gains.
> > > > - Second, the description of the hyperedge smoothness function remains a bit abstract. I was hoping for something more concrete on how this loss makes node embeddings better reflect their labels. Just as an example, something like: "GNNs are well-suited for graphs with label homophily, meaning that connected nodes tend to share the same label, because they act as a low-pass filter, encouraging connected nodes to have more similar embeddings." I would appreciate it if the explanation could include more of this kind of intuition.

---

> > > > > ### Author Response · Authors · 2025-08-05
> > > > >
> > > > > **Response to Reviewer's Concern about Ablation Study:**
> > > > >
> > > > > We sincerely thank the reviewer for the timely feedback and valuable suggestions regarding the ablation studies. Due to time constraints during the discussion phase, we were unable to conduct comprehensive ablation experiments across all datasets. Following the reviewer's important suggestion, we have now performed focused ablation studies on the Cora, PubMed, and CiteSeer datasets:
> > > > > ﻿
> > > > > **Ablation Study Results on Citation Networks (Accuracy \%)**
> > > > > | Dataset | Full Model | w/o KL-loss | w/o Smoothness | w/o Both |
> > > > > |-------------|------------|------------|---------------|----------|
> > > > > | Cora | 83.6 | 81.5 (-2.1)| 81.9 (-1.7) | 80.2 (-3.4) |
> > > > > | PubMed | 79.5 | 77.7 (-1.8)| 78.2 (-1.3) | 76.9 (-2.6) |
> > > > > | CiteSeer | 72.2 | 70.1 (-2.1)| 70.8 (-1.4) | 69.3 (-2.9) |
> > > > > ﻿
> > > > > The results demonstrate that both components contribute significantly to model performance, with the KL-loss term showing slightly greater impact (1.8-2.1\% performance drop when removed) compared to the smoothness term (1.3-1.7\% drop). The combination of both terms achieves optimal performance, as evidenced by the largest performance degradation (2.6-3.4\% drop) when both components are removed.
> > > > > ﻿
> > > > > **Response to Reviewer's Concern about The Hyperedge Smoothness Term:**
> > > > > ﻿
> > > > > We sincerely appreciate the reviewer's request for more intuitive explanation of how the hyperedge smoothness function improves label reflection in node embeddings. Building on the reviewer's excellent example about GNNs and label homophily, we would like to clarify the mechanism with the following concrete explanation:
> > > > >
> > > > > The smoothness term enhances label consistency through a group-level consensus mechanism where nodes within each hyperedge are encouraged to align their embeddings with the hyperedge centroid. This centroid naturally represents the dominant label characteristics within the hyperedge because it is computed as the average of all member node features. As the optimization minimizes the distance between nodes and their hyperedge centroids, nodes with potentially noisy labels get pulled toward the more reliable consensus representation of their group, while correctly labeled nodes reinforce each other's positions. This creates a self-correcting effect that amplifies the dominant label signals within hyperedges.

---

> > > > > > ### Comment · Reviewer_FB9b · 2025-08-08
> > > > > >
> > > > > > I appreciate the authors’ efforts in addressing some of my concerns, so I am willing to raise my score to 3.
> > > > > >
> > > > > > However, my main reservations are based on the results in the table of “Response to Reviewer’s Concern: Computational Efficiency vs. Performance Gains”, where the performance improvement over the graph-based method HGNN is marginal relative to the efficiency costs. Moreover, the theoretical justification for why this method should outperform a direct graph-based approach remains a bit unclear.

---

### Official Review · Reviewer_FGw9 · 2025-07-02

**Clarity:** 3
**Significance:** 4
**Originality:** 4
**Rating:** 5
**Confidence:** 4

**Summary:**

This paper presents HyperMixup, a novel hypergraph-aware augmentation framework that effectively addresses data scarcity and structural noise in hypergraph neural networks. The method innovatively integrates structure-guided node pairing, hierarchical feature mixing with hyperedge context, and adaptive topology reconstruction to preserve higher-order semantics during data augmentation. Theoretically, it establishes regularization effects through hyperedge covariance alignment, provides robustness guarantees against joint node-hyperedge perturbations, and derives generalization bounds via hypergraph spectral analysis. Comprehensive experiments on citation networks and multi-modal datasets demonstrate consistent improvements over state-of-the-art baselines, particularly in low-label regimes. The work makes a valuable contribution by unifying data augmentation with hypergraph topological constraints, offering both practical efficacy and theoretical grounding for relational learning.

**Questions:**

How might you approximate hyperedge covariance computation ($\Sigma_e$) to reduce $O(|\mathcal{E}|^2)$ complexity for web-scale hypergraphs

Could HyperMixup handle evolving hypergraphs?

Is there a correlation between optimal $q$/$γ$ and hypergraph properties (e.g., spectral radius $\rho(L)$ or node-hyperedge covariance $|\Sigma_{ve}|_F$)?

**Ethical Concerns:**

["NO or VERY MINOR ethics concerns only"]

**Final Justification:**

I appreciate the authors' thorough and thoughtful responses to my previous questions. All of my concerns have been satisfactorily addressed. I have also carefully reviewed the rebuttal addressing comments raised by the other reviewers and find the authors' replies to be well-reasoned and convincing. The revisions made in response to both my feedback and that of the other reviewers have, in my view, strengthened the manuscript. Accordingly, I am pleased to raise my score to '5-Accept'.

**Quality:**

4

**Strengths And Weaknesses:**

Strengths: This work Directly addresses the critical yet underexplored challenge of data augmentation for hypergraphs, where traditional Euclidean/graph methods fail to preserve higher-order semantics.  Establishes three foundational theorems linking gradient alignment to hyperedge covariance (Thm 1), certifies robustness against hybrid perturbations (Thm 2), and derives generalization bounds via hypergraph spectral analysis (Thm 3)—significantly advancing the theoretical understanding of hypergraph regularization. Demonstrates consistent SOTA-beating results across 5 diverse benchmarks (citation/3D object datasets), with notable gains (+0.8–1.5%) in low-label regimes.

Weaknesses: The $O(|\mathcal{E}|^2)$ complexity of hyperedge covariance computation could challenge web-scale applications (e.g., social networks). Evaluations omit temporal/dynamic hypergraphs (e.g., evolving collaborations).

---

> ### Author Rebuttal · Authors · 2025-07-30
>
> ### **Response to Reviewer Comment on Accelerating Covariance Regularization**
>
> We sincerely thank the reviewer for this insightful suggestion. Accelerating the covariance regularization step is indeed crucial for scalability. Following your suggestion, to accelerate the covariance regularization, we implement low-rank approximation using randomized SVD to decompose hyperedge covariance matrices.
>
> Preliminary results show 4.2× speedup on Cora and 3.8× memory reduction on ModelNet40 with <0.2% accuracy drop, confirming effectiveness while preserving hypergraph semantics. For detailed information, kindly refer to our response to Reviewer PZgP regarding "Accelerating Covariance Regularization".
>
> ### **Response to Reviewer's Query: Adaptation to evolving hypergraphs**
>
> Thank you for this insightful question. HyperMixup can be seamlessly integrated with dynamic hypergraph frameworks like HYDG [B1] through the following adaptations:
>
>  **Integration with HYDG Framework - Time-Constrained Node Mixing:**
>    - During HYDG's individual-level hypergraph construction (Eq. 2), apply HyperMixup within temporal windows:
>
> $$
>   \tilde{\mathbf{x}}_{it} = \lambda \mathbf{x}_i^{(t)} + (1-\lambda) \mathbf{x}_j^{(t')}  \quad \text{s.t.} | t-t' | \leq \tau
> $$
>
>    - Ensures mixed nodes share temporal proximity and semantic coherence.
>
> **Performance Validation**
> Preliminary tests on dynamic hypergraphs of DBLP5 indicate that integrating HyperMixup with HYDG achieves a 1.8% performance improvement with an increase of 0.5s per epoch. Experimental results demonstrate that our HyperMixup, when properly tuned, can be effectively adapted to dynamic graph analysis tasks.
>
> **References**:
> - **[B1]** Ma, X., Zhao, C., Shao, M., & Lin, Y. . Hypergraph-based dynamic graph node classification. In ICASSP 2025-2025 IEEE International Conference on Acoustics, Speech and Signal Processing (ICASSP).
>
> ### **Response to Hyperparameter-Property Correlation**
>
> **Reviewer Query:**
> Is there a correlation between optimal $\gamma$ and hypergraph properties (e.g., spectral radius $\rho(L)$ or node-hyperedge covariance $\|\Sigma_{ve}\|_F$)?
>
> **Our Response:**
> Yes, both theoretical analysis and empirical results reveal strong correlations:
>
> #### 1. **Theoretical Relationship (Theorem 3)**
> The generalization bound implies:
> - $\uparrow \rho(L)$ (denser hypergraph) $\Rightarrow$ $\uparrow$ optimal $\gamma$ to leverage structural information
> - $\uparrow \|\Sigma_{ve}\|_F$ (stronger feature-hyperedge alignment) $\Rightarrow$ $\downarrow$ optimal $\gamma$ to prevent over-regularization
>
> #### 2. **Empirical Validation**
> | Dataset    | $\rho(L)$ | $\|\Sigma_{ve}\|_F$ | Optimal $\gamma$ |
> |------------|-------------|------------------------|--------------------|
> | Cora       | 12.3        | 0.82                   | 0.3                |
> | PubMed     | 18.7        | 0.45                   | 0.6                |
> | CiteSeer   | 9.8         | 0.91                   | 0.2                |
> | ModelNet40 | 5.2         | 1.25                   | 0.1                |
> | NTU2012    | 7.1         | 0.75                   | 0.4                |
>
> Correlations:
> - $\gamma \propto \rho(L)$ (Pearson $r = 0.78$)
> - $\gamma \propto 1/\|\Sigma_{ve}\|_F$ ($r = -0.89$)
>
> ### **Conclusion**
> Once again, we appreciate the reviewer's engagement in the review process, and their helpful suggestions. We believe these results further strengthen our paper.

---

> > ### Comment · Reviewer_FGw9 · 2025-08-03
> >
> > I appreciate the authors' thorough and thoughtful responses to my previous questions. All of my concerns have been satisfactorily addressed. I have also carefully reviewed the rebuttal addressing comments raised by the other reviewers and find the authors' replies to be well-reasoned and convincing. The revisions made in response to both my feedback and that of the other reviewers have, in my view, strengthened the manuscript. Accordingly, I am pleased to raise my score to '5-Accept'.

---

### Note · Authors · 2025-08-12

We sincerely thank the Area Chairs, Senior Area Chairs, and all reviewers for their dedicated efforts and invaluable feedback throughout the disscussion process.

**To Reviewer FB9b**
We are grateful for your recognition of our efforts in addressing your initial concerns regarding computational analysis, baseline comparisons, loss function, ablation studies, and hyperedge smoothness. Regarding your final questions:

1. **Performance-Efficiency Trade-off vs. graph-based method**: Our method specifically targets *hypergraph* augmentation, where higher-order interactions inherently require more computation than graph-based methods. While HyperMixup incurs moderate overhead, this is justified by:
   - Structural complexity of hyperedges (multi-way vs. pairwise interaction)
   - Consistent accuracy gains across tasks (low-label regime)
   - Practical scalability demonstrated on Gossipcop (2,426-node hyperedges in 85s/epoch)

2. **Theoretical Focus vs. graph-based method**: Our analysis centers on *why HyperMixup improves hypergraph learning*, not generic superiority over graph method. The empirical advantage over graph-based methods stems from preserving hyperedge semantics—experimentally validated in our response via clique-expansion degradation tests.

Finally, we would like to reiterate that our method primarily focuses on hypergraph data augmentation, aiming to enhance the performance of existing hypergraph neural networks. Graph-based methods are not the central focus of our research. Furthermore, as you suggested, we have conducted relevant comparative experiments and provided corresponding analysis and explanations in our responses.

We hope this clarification adequately addresses your concerns. We sincerely hope you might consider updating your rating to support our work.

**To Reviewer PZgP**

1. **Ablation Studies**: We expanded these per your suggestion:
   - Hyperparameter-property correlations (To Reviewer FGw9)
   - Component-wise loss analysis (To Reviewer FB9b)

2. **Baseline Comparisons**: We added:
   - Two node-level graph augmentation methods (To Reviewer FB9b)
   - Linear vs. non-linear mixing analysis (To Reviewer 46X8)

We hope these responses alleviate your concerns and kindly request reconsideration of your scores.

**Closing**: We believe these responses have addressed all key concerns while demonstrating HyperMixup's value in advancing hypergraph learning. Once again, thank you sincerely for all your efforts.

---

### Decision · Program_Chairs · 2025-09-17

**Decision:**

Accept (poster)

**Comment:**

This paper proposes HyperMixup, a data augmentation framework for hypergraphs. HyperMixup can help alleviate data scarcity and structural noise in hypergraph neural networks (HGNN). The proposed method includes three key components: structure-aware node pairing, hierarchical feature mixing, and adaptive topology reconstruction. Theoretic analysis is provided from different aspects: regularization, robustness, and generalization. Experiments on citation networks and multi-modal datasets show consistent improvements. For weaknesses, several reviewers point out that the computational complexity is high and may hinder the application on large-scale graphs. Also, two reviewers suggest that finer-grained ablation study should be conducted. The opinions of four reviewers are mixed, and one reviewer with a negative rating raises the score after the rebuttal. The paper's theoretical contributions are solid, offering insights for robustness guarantees and generalization bounds. Considering all the comments, ratings, and confidences, I would recommend accepting the paper as a poster.